# CD1d-Restricted NKT Cells Promote Central Memory CD8^+^ T Cell Formation via an IL-15-pSTAT5-Eomes Axis in a Pathogen-Exposed Environment

**DOI:** 10.3390/ijms26157272

**Published:** 2025-07-28

**Authors:** Yingyu Qin, Yilin Qian, Jingli Zhang, Shengqiu Liu

**Affiliations:** 1Jiangsu Provincial Key Laboratory of Critical Care Medicine, Department of Pathogenic Biology and Immunology, School of Medicine, Southeast University, Nanjing 210096, China; qyl18305814880@163.com (Y.Q.); jinglizhang_2004@163.com (J.Z.); 2Department of Pathogenic Biology and Immunology, School of Medicine, Southeast University, Nanjing 210096, China; 15251815357@163.com

**Keywords:** CD1d-restricted NKT, central memory CD8^+^ T, IL-15, Eomes

## Abstract

The generation of memory CD8^+^ T cells is essential for establishing protective T cell immunity against pathogens and cancers. However, the cellular and molecular mechanisms underlying memory CD8^+^ T cell formation remain incompletely understood. Reliance on specific pathogen-free (SPF) models, characterized by restricted microbial exposure, may limit our understanding of physiologically relevant immune memory development. This study reveals that CD1d-restricted NKT cells regulate central memory T cell (TCM) generation exclusively in a microbe-rich (“dirty”) environment. Under non-SPF housing, CD1d^+^/^−^ and Ja18^+^/^−^ mice exhibited enhanced TCM formation compared to NKT-deficient controls (CD1d^−^/^−^/Ja18^−^/^−^), demonstrating that microbial experience is required for NKT-mediated TCM regulation. Mechanistically, CD1d-restricted NKT cells increased IL-15Rα expression on CD4^+^ T cells in CD1d^+^/^−^ mice, potentiating IL-15 trans-presentation and thereby activating the IL-15/pSTAT5/Eomes axis critical for TCM maintenance. Functional validation through adoptive transfer of CFSE-labeled OT-1 memory cells revealed an NKT cell-dependent survival advantage in CD1d^+^/^−^ hosts. This provides direct evidence that microbiota-experienced niches shape immune memory. Collectively, these findings establish CD1d-restricted NKT cells as physiological regulators of TCM generation and suggest their potential utility as vaccine adjuvants to enhance protective immunity.

## 1. Introduction

CD8^+^ T cells are a key type of lymphocytes in the adaptive immune system, which plays a critical role in combating and controlling malignancies and intracellular infections. Once naïve CD8^+^ T cells are activated, a largely autonomous program of proliferation and differentiation is induced, which results in effector and memory CD8^+^ T cell generation [1]. Memory CD8^+^ T cells are the most potent subset of cells in the defense against infections and tumors; they possess long-term survivability and can undergo proliferation and acquire effector function rapidly and robustly upon antigen re-exposure [2]. Thus, understanding the factors driving efficient memory CD8^+^ T cell generation is a major goal with a view to improving vaccine and immunotherapy design.

It is important to realize that most studies have been conducted in mice housed in specific pathogen-free (SPF) housing, aiming to reveal the fundamental principles of the generation and maintenance of T-cell memory and to deduce how these findings may translate to humans. However, the immune system of humans is naturally a pathogen-rich environment. Some studies have suggested that the immune system of “dirty” mice, which are exposed to multiple mouse pathogens, is more similar to that observed in adult humans [3,4,5]. Therefore, “dirty” mice can be used to better define the role of the adaptive immune system in the generation and maintenance of memory T cells within innate immune cells.

It is believed that helper cells are important for promoting effective cellular immunity by enhancing CD8^+^ T cell clonal expansion, differentiation and survival. Besides CD4^+^ T helper cells, a large body of studies have determined that type I Natural killer T (NKT) cells, also known as invariant NKT cells, can also achieve this task [6]. NKT cells are a lymphocyte lineage belonging to the innate immune system; specifically, they recognize lipid antigens presented by CD1d molecules expressed by antigen-presenting cells [7,8]. As an innate type of lymphocyte, NKT cell response is immediate and robust. Upon stimulation, NKT cells rapidly produce large amounts of T-helper (Th)-1 and Th2-type cytokines, which induce the subsequent activation of other types of immune cells, including NK cells, DCs and T and B cells [9,10]. α-galactosylceramide (αGalCer) is a synthetic antigen that can potently and selectively activate iNKT cells [11]. Several studies have shown that iNKT cells stimulated by α-galactosylceramide (αGalCer) may promote memory CD8^+^ T cell differentiation [12,13,14].

However, NKT cells activated by synthetic lipid antigens cannot fully represent the true physiological functions of NKT cells, as these compounds induce non-physiological hyperactivation and cytokine profiles. Importantly, physiological iNKT activation occurs via microbial-triggered cytokine receptors (e.g., IL-12R/IL-18R) [15], highlighting the necessity of pathogen exposure for studying their endogenous functions. Our prior work showed iNKT cells enhance CD8^+^ T cell effector function without exogenous antigens [16], yet their role in memory differentiation under physiologically relevant (microbe-rich) conditions remains unknown.

In this study, we resolved this gap using non-SPF (“dirty”) housing to model natural microbial exposure. We demonstrate that CD1d-restricted NKT cells drive central memory T cell (TCM) formation exclusively in a microbe-rich environment—a dependency absent in SPF settings. Eomesodermin (Eomes) is a transcription factor crucial for the development and function of CD8^+^ memory T cells [17]. In addition, IL-15 contributes to the homeostatic proliferation and survival of memory CD8^+^ T cells under steady-state conditions [18]. Mechanistically, we observed that Eomes expression is strictly correlated with TCM formation. CD1d^+^/^−^ mice exhibited significantly elevated IL-15 (mRNA and protein) levels and increased IL-15Rα expression on CD4^+^ T cells compared to CD1d^−^/^−^ mice, demonstrating that CD1d-restricted NKT cells potentiate IL-15 trans-presentation capability, likely mediated by CD4^+^ T cells. Subsequent investigations established that CD1d-restricted NKT cells promote TCM formation through an IL-15-STAT5 phosphorylation–Eomes upregulation axis. Adoptive transfer experiments confirmed an NKT-dependent survival advantage for memory CD8^+^ T cells specifically in microbiota-experienced hosts.

## 2. Results

### 2.1. CD1d-Restricted NKT Cells Promote TCM Formation in Steady State

To investigate the physiological role of NKT cells in memory CD8^+^ T cell generation, we compared littermate CD1d^+/−^ and CD1d^−/−^ mice (lacking NKT cells) housed under non-SPF conditions. Without experimental intervention, both strains exhibited an age-dependent increase in central memory T cell (TCM) frequencies, but CD1d^+/−^ mice consistently showed higher TCM frequencies than CD1d^−/−^ mice from early to late ages (Figure 1A,B). To determine whether CD1d-restricted NKT cells promote TCM formation specifically in a pathogen-exposed environment, we compared littermate CD1d^+/−^ and CD1d^−/−^ mice housed under SPF conditions. Interestingly, CD1d^+/−^ mice showed decreased TCM (CD44^+^ CD62L^+^) and TEM (CD44^−^ CD62L^+^) populations compared to CD1d^−/−^ mice, which was opposite to the results under antigen exposure conditions (Figure 1C). Since CD1d deficiency depletes all CD1d-restricted NKT cells, we further assessed Jα18^−/−^ mice, which lack only iNKT cells. Consistent with our earlier findings, Jα18^−/−^ mice had reduced TCM populations compared to Jα18^+/−^ controls across all ages (Figure 1D). However, the complete ablation of all CD1d-restricted NKT cells (CD1d^−/−^) led to more severe impairments in the formation of central memory T cells (TCMs) compared to the ablation of iNKT cells alone (Jα18^−/−^). Finally, a rescued experiment was processed using adoptive NKT cell transfer into CD1d^−/−^ mice. Compared to the non-cell-transfer group, slightly elevated frequencies of TCM were observed in the peripheral blood and inguinal lymphoid tissues in the NKT cell transfer group (Figure 1E). The results indicate that NKT cells promoting CD8^+^ TCM formation are not crucially dependent on CD1d molecules. Collectively, these results indicate that CD1d-restricted NKT cells promote TCM formation under non-SPF conditions.

### 2.2. CD1d-Restricted NKT Cells Promote Homeostatic Proliferation of Memory CD8^+^T Cells

To determine whether CD1d-restricted NKT cells regulate memory T cell survival and homeostatic proliferation, we transferred CFSE-labeled memory OT-1 cells into CD1d^−/−^ and CD1d^+/−^ littermates (Figure 2A). Peripheral blood (PBL) analysis on days 7, 21 and 35 post-transfer revealed a progressive decline in OT-1 cell numbers, with CD1d^+/−^ mice consistently retaining more OT-1 cells than CD1d^−/−^ mice (Figure 2B). On day 35, we also observed higher numbers of memory OT-1 cells in the spleen and inguinal lymph nodes (ILNs) of CD1d^+/−^ mice (Figure 2C). To assess homeostatic proliferation, we analyzed CFSE dilution in transferred OT-1 cells. CD1d^+/−^ mice exhibited significantly more OT-1 cells in division cycles D2 and D3 compared to CD1d^−/−^ mice (Figure 2D). Together, these results indicate that CD1d-restricted NKT cells enhance memory T cell survival and homeostatic proliferation.

### 2.3. Eomes Expression Is Strictly Correlated with TCM Formation

Underlying CD8^+^ T cell differentiation and function are regulated by transcription factors. T cell factor-1 (TCF-1) (encoded by the TCF7 gene) is a transcription factor that plays an important role during the formation and maintenance of CD8^+^ T cell memory in acute infections [19]. The T-box transcription factors T-box expressed in T cells (T-bet) and eomesodermin (Eomes) have been implicated as master regulators of CD8 T cell differentiation and function [20]. T-bet is associated with effector function and terminal effector T cell subsets and is also expressed in memory T cells [21]. Eomes is primarily associated with memory T cell formation [22]. We then examined whether these transcription factors are involved in the generation of TCM mediated by CD1d-restricted NKT cells. We found CD1d-restricted NKT cells slightly enhanced the frequencies of TCF1^+^ CD62L^+^ (Figure 3A) and T-bet^+^ CD62L^+^ population (Figure 3B). However, CD1d-restricted NKT cells significantly enhanced the frequencies of Eomes^+^CD62L^+^ populations (Figure 3C). Furthermore, we found that the populations of TCF1^+^ CD62L^+^, T-bet^+^ CD62L^+^ and Eomes^+^ CD62L^+^ were positively correlated with the proportion of TCM. Notably, among these correlations, the association between Eomes and TCM was the strongest (Pearson r = 0.9745, *p* < 0.0001) (Figure 3D), which implies that Eomes may play a role in TCM generation mediated by CD1d-restricted NKT cells.

### 2.4. iNKT Cells Promote the Homeostatic Proliferation of Memory OT-1 Cells

IL-15 and IL-7 are critical cytokines for maintaining homeostatic proliferation and survival of memory CD8^+^ T cells under steady-state conditions [18], while IL-21, a hallmark cytokine of follicular helper T (TFH) cells, drives the differentiation of central and effector memory CD8+ T cells [23,24]. To investigate potential regulatory differences between CD1d^+/−^ and CD1d^−/−^ mice, we compared splenic cytokine gene expression profiles. Although IL-7 and IL-21 mRNA levels showed no significant differences between genotypes, CD1d^+/−^ mice exhibited markedly elevated IL-15 mRNA expression compared to their CD1d^−/−^ counterparts (Figure 4A). This transcription difference was corroborated at the protein level, with higher serum IL-15 concentrations observed in CD1d^+/−^ mice (Figure 4B).

Then we identified the source cells responsible for IL-15 production. Given that secreted IL-15 binds to transmembrane IL-15Rα with high affinity and is presented on the cell surface, where it acts in trans on adjacent IL-2Rβ:γ-expressing cells [25], we examined IL-15Rα surface expression to evaluate the IL-15 trans-presentation capacity of the determined cells.

Given the absence of NKT cells in CD1d^−/−^ mice and our previous observation that NKT cells do not promote TCM formation under SPF conditions (Figure 1C), we examined whether environmental factors influence NKT cell production of IL-15 to regulate the generation of TCM. We therefore compared IL-15Rα surface expression on wild-type NKT cells under conventional (non-SPF) versus SPF housing conditions. While NKT cells demonstrated baseline IL-15 production capacity, no significant environmental differences were detected in this population (Figure 4C). Intriguingly, T cells from mice housed in a “dirty” environment displayed elevated IL-15Rα levels (Figure 4C), suggesting potential NKT cell-mediated regulation of T cell IL-15 production. We subsequently performed a comparison of IL-15Rα expression levels on CD4^+^ T and CD8^+^ T cells between CD1d^+/−^ and CD1d^−/−^ mice. Notably, CD1d^+/−^ mice exhibited significantly higher IL-15Rα levels on CD4^+^ T helper cells compared to CD1d^−/−^ mice, whereas CD8^+^ T cells showed comparable receptor expression in both groups (Figure 4D). Macrophages and DCs are also IL-15-producing cells [26]. However, the level of IL-15Rα was comparable between CD1d^+/−^ and CD1d^−/−^ mice (Figure 4E). Collectively, these findings suggest that CD1d-restricted NKT cells contribute to an immunological milieu characterized by enhanced IL-15 availability and augmented potential for IL-15 trans-presentation, particularly by CD4^+^ T cells due to their heightened IL-15Rα expression, which potentially contributes to an enhanced survival and homeostatic proliferation of CD8^+^ TCM cells in CD1d^−/−^ mice.

### 2.5. CD1d-Restricted NKT Cells Mediate Homeostatic Formation of TCM Through IL-15-pSTAT5-Eomes Axis

Our data demonstrate that CD1d-restricted NKT cells significantly increase the frequency of Eomes^+^CD62L^+^ CD8^+^ T cells (Figure 3C), with Eomes expression levels strongly correlating with the proportion of central memory T (TCM) cells (Figure 3D). Notably, TCM compartments in the presence of CD1d-restricted NKT cells exhibited elevated Eomes expression (Figure 5A). Several reports have addressed that stimulation-mediated by IL-15 through CD122 leads to the induction of Eomes expression [26,27,28,29,30]. Therefore, we postulated a functional link between NKT cell-mediated IL-15 signaling and Eomes regulation. Consistent with this hypothesis, we observed significantly increased phosphorylation of STAT5 (pSTAT5)—a downstream effector of IL-15 signaling triggered by IL-15Rβ:γ engagement—in TCM cells from mice with CD1d-restricted NKT cells (Figure 5B). Furthermore, CD1d-restricted NKT cells markedly enhanced Eomes expression specifically within the CD62L^+^ CD122^+^ T cell subset (Figure 5C). Together, these findings establish that CD1d-restricted NKT cells drive TCM generation, potentially through an IL-15-pSTAT5-Eomes signaling axis.

## 3. Discussion

The differentiation and maintenance of memory CD8^+^ T cells represent critical determinants of long-term immunological protection against pathogens and malignancies. While prior studies have established the importance of helper cells in shaping CD8+ T cell memory, the present work provides novel insights into the physiological role of CD1d-restricted NKT cells in TCM formation under conventional (non-SPF) housing conditions. Our findings demonstrate that CD1d-restricted NKT cells, particularly iNKT cells, are essential for sustaining TCM populations through an IL-15-dependent mechanism involving STAT5 phosphorylation and Eomes upregulation. These observations extend current paradigms by highlighting how environmental pathogen exposure modulates innate–adaptive crosstalk to influence memory T cell biology.

A striking feature of this work is the environmental specificity of NKT cell function. Unlike SPF-housed mice, in which NKT cells showed a negative influence on TCM frequencies, conventionally housed (“dirty”) mice exhibited a pronounced reliance on CD1d-restricted NKT cells for TCM maintenance. This dichotomy underscores how microbial exposure primes innate lymphocytes to acquire regulatory roles in adaptive immunity—a concept increasingly recognized in mucosal tissues but less explored systemically. For instance, commensal microbiota educate group 3 innate lymphoid cells (ILC3s) to support intestinal T cell memory [31]. Here, we extend this paradigm by demonstrating that systemic pathogen exposure licenses NKT cells to orchestrate CD8^+^ TCM homeostasis, likely via tonic activation through cytokine receptors (e.g., IL-12R, IL-18R) or indirect microbial ligand engagement. Such a mechanism positions NKT cells as environmental rheostats, dynamically calibrating memory T cell reserves to match historical pathogen encounters. This aligns with evolutionary demands for immune systems to optimize resource allocation: in pathogen-rich settings, sustained NKT-mediated TCM support ensures rapid recall responses, whereas sterile environments may prioritize effector readiness over memory longevity.

We found that TCM damage was more pronounced in CD1d^−/−^ mice than in Jα18^−/−^ mice, suggesting that both invariant NKT (iNKT) cells and various CD1d-restricted NKT cell subsets, as well as certain γδ T cell subsets that have been reported to engage CD1d-restricted lipid antigen presentation, contribute functionally, potentially through distinct cytokine microenvironments. These findings align with emerging evidence supporting NKT cell heterogeneity in tissue-specific immunity [32].

Eomes emerged as the dominant transcription correlate of NKT-mediated TCM maintenance, outperforming TCF1 and T-bet in predictive strength. This aligns with Eomes’ dual role in promoting memory precursor survival and metabolic fitness [33,34]. IL-15 signaling via STAT5 phosphorylates and stabilizes Eomes, enabling TCM cells to compete for survival signals in IL-15-limited niches. Notably, NKT cells selectively boosted Eomes expression in CD62L^+^ CD122^+^ TCM subsets, suggesting a focused effect on self-renewing memory populations. This specificity may explain the superior homeostatic proliferation of transferred OT-1 cells in CD1d^+/−^ hosts, as Eomes promote T cell survival through enhancement of mitochondrial fitness and adaptability to cytokine fluctuations [33,34,35]. Mechanistically, the IL-15-STAT5-Eomes axis likely synergizes with TCF1 to enforce memory stemness while counteracting T-bet-driven effector differentiation, thereby stabilizing the TCM phenotype. The IL-15-Eomes axis identified in this study complements established pathways for memory T cell maintenance [27,28,29,34] and positions NKT cells as novel regulators within this cytokine network. Notably, the environmental dependency of this phenomenon—absent in specific pathogen-free (SPF) reared mice—underscores the critical role of microbial exposure in enabling NKT cells to regulate memory T cell function, possibly via sustained activation of pattern recognition receptors [36].

While this study advances our understanding of NKT cell biology, several limitations warrant consideration. For instance, the precise crosstalk between NKT and CD4^+^ T cells, as well as the spatial dynamics of IL-15, remain unvalidated (e.g., through IL-15 knockout models). Despite these gaps, they do not diminish the central role of NKT cells in maintaining TCM via the IL-15/Eomes axis. Further mechanistic validation would reinforce, but not negate, the proposed model.

## 4. Materials and Method

### 4.1. Mice

B6(Cg)-Traj18^tm1.1Kro^/J (Jα18^−/−^) mice [37] were kindly provided by Pro. Xinzhi Wang (China Pharmaceutical University, Nanjing, China) and B6.129S6-Del (3Cd1d2-Cd1d1) 1Sbp/J (CD1d^−/−^) mice were purchased from Jackson laboratories and kindly provided by Pro. Zhigang Lei (Nanjing Medical University, Nanjing, China). The littermates of CD1d^+/−^ and CD1d^−/−^ mice were generated through the crossing of CD1d^+/−^ and CD1d^−/−^ mice. The same applies to Jα18^+/−^ and Jα18^−/−^ mice. The mice that were not highlighted as housed in SPF were housed in open cages in conventional animal facilities of Southeast University, which do not have air cleanliness standards, but they maintain regular ventilation (8 times/hour) and control room temperature within the range of 24 °C, in accordance with the Chinese standard for laboratory animal environment and housing facilities (GB 14925-2023). Ovalbumin (OVA)-specific TCR transgenic mice (OT-I) were purchased from Aniphe BioLab (Nanjing, China), and these mice were kept in specific pathogen-free animal facilities of Southeast University. All animal experiments were approved by institutional guidelines established by the Committee of Ethics on Animal Experiments of Southeast University.

### 4.2. Flow Cytometric Analysis

Lymphocytes were analyzed with following antibodies: CD3-PE/Percp5.5 (17A2), CD8α-Percp5.5/FITC (53-6.7), CD44-APC (IM7), CD62L-APC.Cy7/BV605 (MEL-14), TCRVβ5.1,5.2-APC (MR9-4), CD45.1-PE/BV786 (A20), TCRVα2-PE (B20.1), NK1.1-FITC (S17016D), IL-15Rα-APC (6B4C88), CD122-FITC (TM-β1), pSTAT5-PE (A17016B.Rec), TCF1-BV421 (S33-966), Eomes-PE (W17001A), T-bet-PE.Cy7 (4B10). LIVE/DEAD Fixable Blue viability-BV510. For transcription factor staining, the lymphocytes underwent intracellular staining using a Fixation/Permeabilization Solution Kit (BD Pharmingen, San Diego, CA, USA) after surface staining. For IL-15Rα staining, the lymphocytes were incubated in culture medium and treated with PMA/ionomycin for 4 h followed by surface staining. For phosphorylated STAT5 detection, the splenocytes were stimulated with 50 ng/mL IL-15 at 37 °C for 30 min, followed by staining of pSTAT5 and surface staining using a standard protocol [38]. Data acquisition was performed on a five-laser BD LSRFortessa™ Cell Analyzer Flow Cytometer.

### 4.3. Real-Time Quantitative PCR

The spleens were isolated from indicated mice, followed by TRIzol treatment and subsequent RNA extraction. Quantitative real-time PCR (qPCR) was performed with the use of SYBR Green (Vazyme, Nanjing, China) on a QuantStudio™ 3 Real-Time PCR System (Thermo fisher, Waltham, MA, USA). All reactions were run in triplicate. Primers for real-time qPCR are as follows: IL-7: F: 5′-CAGGAACTGATAGTAATTGCCCG-3′ and R: 5′-CTTCAACTTGCGAGCAGCACGA-3′; IL-15: F: 5′-GTAGGTCTCCCTAAAACAGAGGC-3′ and R: 5′-TCCAGGAGAAAGCAGTTCATTGC-3′; IL-21: F: 5′-CGCCTCCTGATTAGACTTCG-3′ and R:5′-TGGGTGTCCTTTTCTCATACG-3′; β-actin: F: 5′-GCCAACCGTGAAAAGATGACCCAG-3′ and R: 5′-ACGACCAGAGGCATACAGGGACAG-3′.

### 4.4. IL-15 Serum ELISA

About 200 μL of peripheral blood was collected from the littermates of CD1d^+/−^ and CD1d^−/−^ mice. The samples were incubated at room temperature for 30min, followed by centrifugation at 2000× *g* for 20 min to remove the clot. The concentration of IL-15 in the serum was subsequently measured using an IL-15 ELISA kit (ELK Biotechnology, Wuhan, China).

### 4.5. Adoptive Transfer Strategy

For memory OT-1 cell transfer, OT-1 mice were immunized with 100 μg of OVA_257–264_ peptides emulsified in Freund’s complete adjuvant (CFA, Sigma-Aldrich, St. Louis, MO, USA). Following one month of immunization, the memory differentiation of OT-1 cells was examined through the expression of CD44 and CD62L using FACS analysis. The total population of memory OT-1 T cells was isolated from the spleen and inguinal lymph nodes (ILNs) via a negative CD8a^+^ T Cell Isolation Kit (Miltenyi Biotec, Shanghai, China, 130-104-075) according to the manufacturer’s instructions; subsequently, memory CD8^+^T cells were labeled with 5 μM carboxyfluorescein succinimidyl ester (CFSE). Subsequently, 6 × 10^5^ CFSE-labeled memory OT-1 cells were intravenously injected into CD1d^+/−^ and CD1d^−/−^ mice.

For NKT cell transfer, NKT cells were isolated from splenocytes and liver lymphocytes that were enriched via gradient centrifugation (40% Percoll, 600× *g*, 15 min, RT) from CD1d^+/−^ mice using the NK1.1^+^ iNKT cell isolation kit (Milteny Biotec, # 130-096-513). Subsequently, 4 × 10^5^ isolated NKT cells were intravenously injected into CD1d^−/−^ mice.

### 4.6. Statistical Analysis

Statistical analyses were performed using GraphPad Prism (version 8). Unpaired Student’s *t*-test was employed to determine the differences between two data sets. One-way analysis of variance (ANOVA) was used for more than 2 comparison groups. The significance levels considered were *p* < 0.05 (*), *p* < 0.01 (**), *p* < 0.001 (***), and *p* < 0.0001 (****). The mean ± standard error of the mean (SEM) was used to express the data.

## 5. Conclusions

This study demonstrates that CD1d-restricted NKT cells serve as pivotal regulators of the homeostasis of CD8^+^ central memory T cells (TCM) in hosts exposed to pathogens. By translating microbial exposure into IL-15 signaling-driven, Eomes-dependent memory maintenance, these cells link innate environmental perception with adaptive immune memory. This homeostatic function complements their acute helper role, underscoring the multifaceted nature of NKT cells. Future investigations into the temporal and spatial regulation of microbial licensing signals and IL-15 delivery could refine strategies for vaccine development and T cell-based therapies while emphasizing the profound impact of environmental context on the shaping of immune memory.

## Figures and Tables

**Figure 1 ijms-26-07272-f001:**
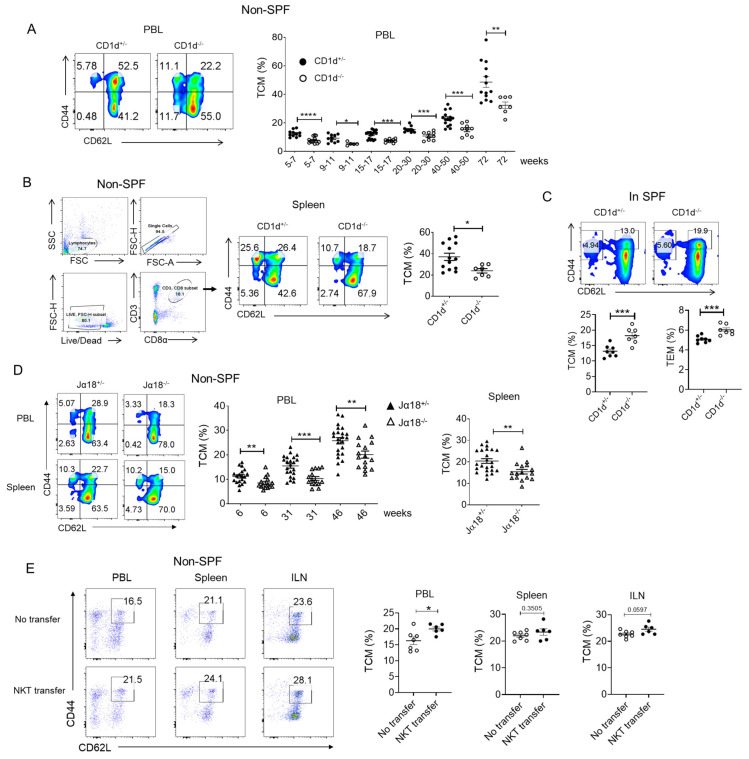
Comparison of the TCM population in littermates of CD1d^+/−^ (Jα18^+/−^) and CD1d^−/−^ (Jα18^−/−^) mice under steady-state conditions. (**A**) Representative flow cytometry plots showing the analysis of central memory (CD44^+^ CD62L^+^) CD8^+^ T cells (TCM) in the peripheral blood (PBL) (left panel). Statistical data representing the percentages of TCM in littermates of CD1d^+/−^ and CD1d^−/−^ mice at various ages (right panel). (**B**) Representative flow cytometry plots showing the analysis of TCM in the spleens of 72-week-old CD1d^+/−^ and CD1d^−/−^ mice, along with their gating strategy (left panel). Statistical data depicting the percentages of TCM in littermates of CD1d+/− and CD1d^−/−^ mice (right panel). (**C**) Representative flow cytometry plots showing the analysis of TCM and TEM in the PBL of SPF mice. Statistical data representing the percentages of TCM and TEM in littermates of CD1d^+/−^ and CD1d^−/−^ mice at 8–10 weeks (right panel). (**D**) Representative flow cytometry plots showing the analysis of TCM in the PBL and spleens (46-week-old mice) (left panel). Statistical data representing the percentages of TCM in littermates of Jα18^+/−^ and Jα18^−/−^ mice at various ages (right panel). (**E**) A total of 4 × 10^5^ NKT cells isolated from CD1d^+/−^ mice were transferred into CD1d^−/−^ mice. After 4 weeks of transfer, the frequencies of CD8^+^ TCM in the PBL, spleens and inguinal lymph nodes (ILNs) were analyzed. Non-cell-transfer group (*n* = 7); NKT cell transfer group (*n* = 6). The number of CD1d^+/−^ and CD1d^−/−^ mice in conventional condition (non-SPF): 5–7 weeks, *n* = 11/13; 9–11 weeks, *n* = 10/5; 15–17 weeks, *n* = 18/10; 20–30 weeks, *n* = 11/9; 40–50 weeks, *n* = 15/9; 72 weeks, *n* = 13/7. The number of CD1d^+/−^ and CD1d^−/−^ mice in SPF: 8–10 weeks, *n* = 8/7. The number of Jα18^+/−^ and Jα18^−/−^ mice: 6 weeks, *n* = 19/15; 31 weeks, *n* = 22/15; 46 weeks, *n* = 23/15. Means with SEM are shown. Statistics: Statistical significance was determined using one-way ANOVA with Tukey tests for multiple-group comparisons (**A**,**C**); non-paired two-tailed Student’s test (**B**,**D**,**E**). * *p* < 0.05, ** *p* < 0.01, *** *p* < 0.001, **** *p* < 0.0001.

**Figure 2 ijms-26-07272-f002:**
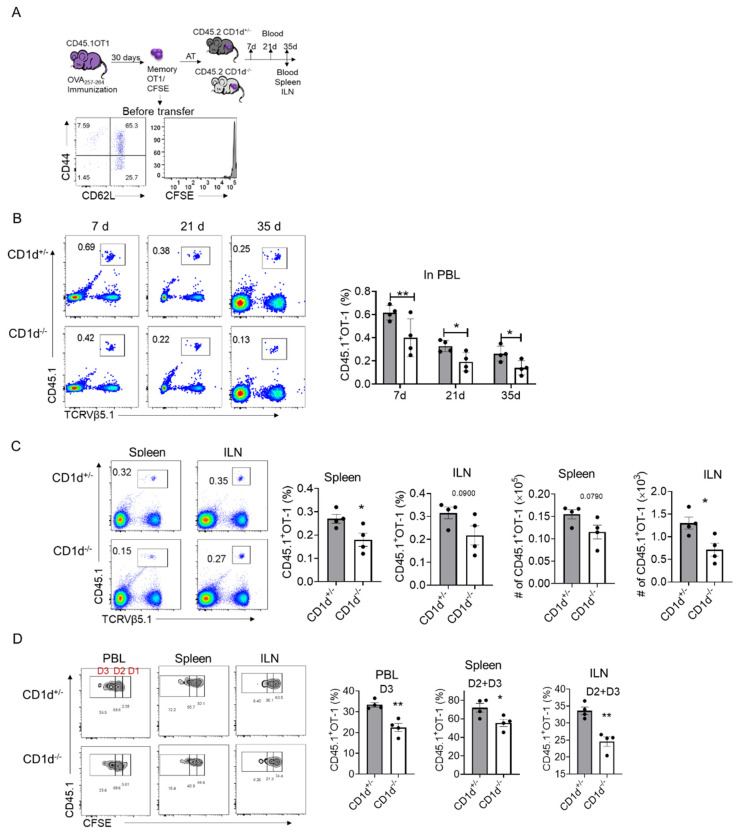
CD1d−restricted NKT cells promote the survival and homeostatic proliferation of memory OT-1 cells. (**A**–**D**) A total of 6 × 10^5^ CFSE-labeled memory CD45.1^+^ OT-1 cells were intravenously injected into CD45.2^+^ CD1d^+/−^ and CD45.2^+^ CD1d^−/−^ mice, respectively. Four mice per group. The schematic diagram of the experimental workflow and flow cytometry plots illustrate the memory phenotypes of OT-1 cells following CFSE labeling prior to transfer. The transferred OT-1 cells were detected in PBL at the indicated time points. The transferred OT-1 cells were detected in the spleens and inguinal lymph nodes (ILNs) on day 35. On day 35, the frequencies of OT-1 cells in CFSE division zones (D2, D2 + D3) were examined in PBL, spleens and ILNs. Each dot represents a mouse. CD1d^+/−^ (*n* = 4); CD1d^−/−^ (*n* = 4). Representative results are from three independent experiments with consistent results. Means with SEM are shown. Statistics: non-paired two-tailed Student’s test: * *p* < 0.05, ** *p* < 0.01.

**Figure 3 ijms-26-07272-f003:**
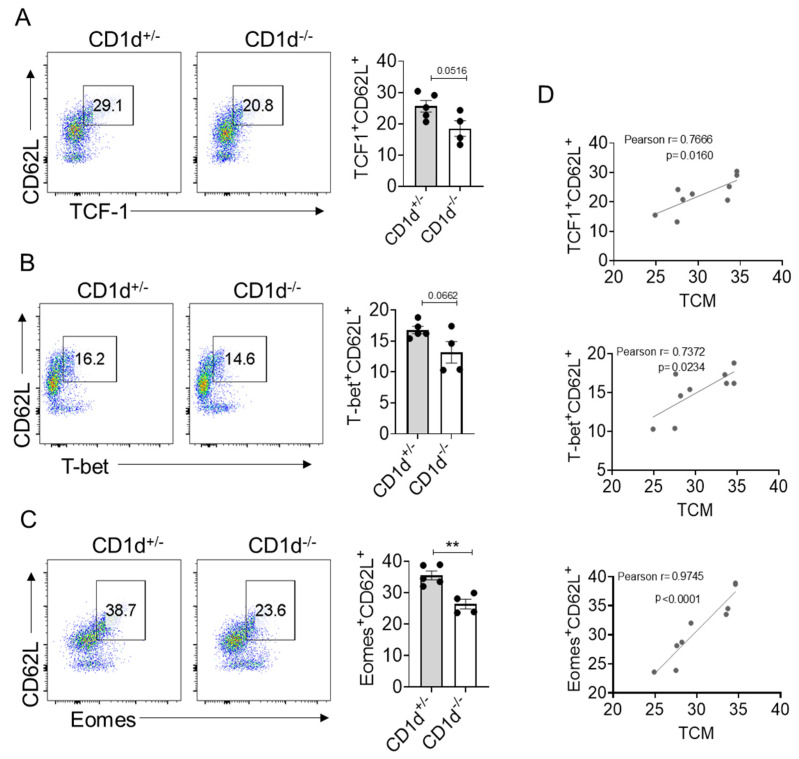
Examination of transcription factors related to TCM. The proportions of TCF1^+^ CD62L^+^ (**A**), T-bet^+^CD62L^+^ (**B**) and Eomes^+^ CD62L^+^ (**C**) within CD8^+^ T cells were examined via FACS analysis in the spleens of 10–12-week-old CD1d^+/−^ and CD1d^−/−^ mice. The frequencies of TCF1^+^ CD62L^+^, T-bet^+^ CD62L^+^, and Eomes^+^ CD62L^+^ CD8^+^T cells were correlated with CD44^+^ CD62L^+^ CD8^+^ T cells (TCM) (**D**). Each dot represents a mouse. CD1d^+/−^ (*n* = 5); CD1d^−/−^ (*n* = 4). Representative results are from three independent experiments with consistent results. Data are presented as means ± SEM. Statistical significance was measured by unpaired two-tailed Student’s *t* test (**A**–**C**) and Pearson’s correlation coefficient. ** *p* < 0.01.

**Figure 4 ijms-26-07272-f004:**
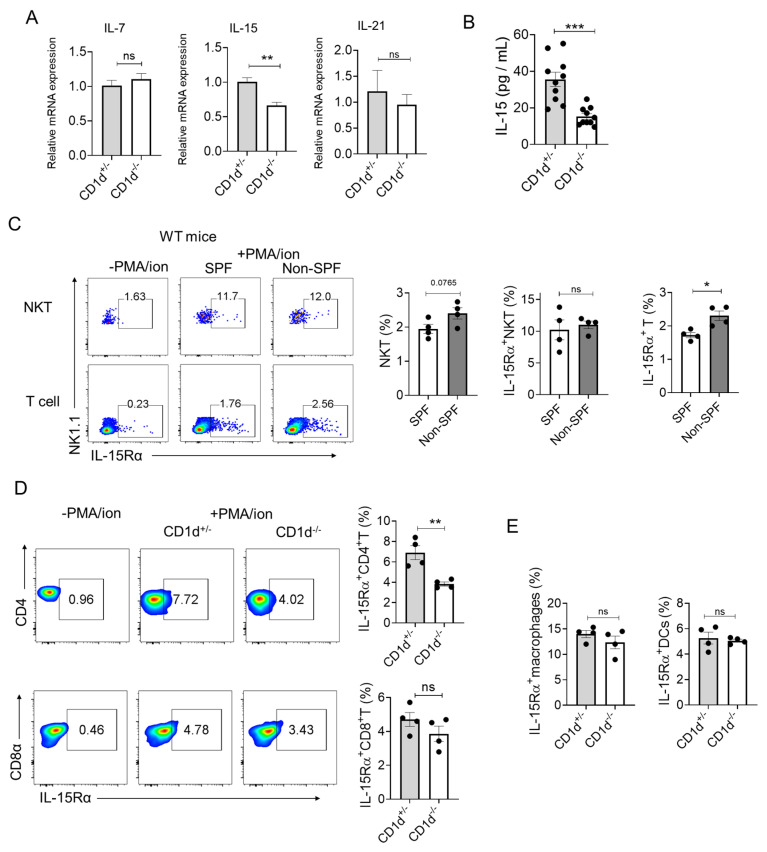
Examination of cytokines related to TCM formation. (**A**) qPCR analysis of the mRNA level of IL-7, IL-15, and IL-21 in freshly isolated splenocytes from 10–12-week old CD1d^−/−^ and CD1d^+/−^ mice. The data were collected from CD1d^+/−^ (*n* = 9) and CD1d^−/−^ (*n* = 8) mice. (**B**) The protein levels of IL-15 were detected in the serum of CD1d^+/−^ and CD1d^−/−^ mice, with 10 mice in each group. (**C**) The expression levels of IL-15Rα on the surface of NKT or T cells in the spleens of wild-type mice bred in SPF or “dirty” conditions were examined. Four mice per group. The data are representative of three independent experiments. The expression levels of IL-15Rα on the surface of CD4^+^ T cells and CD8^+^ T cells (**D**), Macrophages and DCs (**E**) in the spleens of CD1d^+/−^ and CD1d^−/−^ mice were compared. Four mice per group. The data are representative of three independent experiments. Means with SEM are shown. Statistics: non-paired two-tailed Student’s test: * *p* < 0.05, ** *p* < 0.01, *** *p* < 0.001, ns *p* > 0.05.

**Figure 5 ijms-26-07272-f005:**
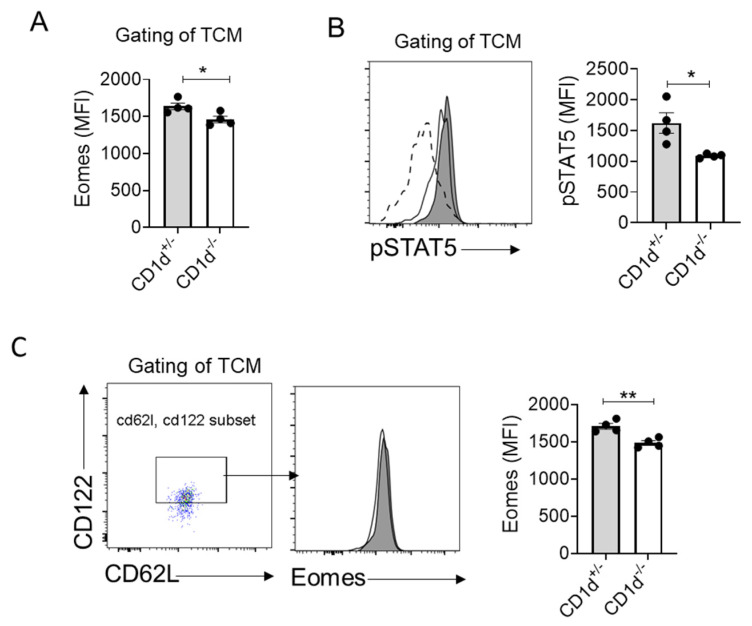
CD1d-restricted NKT cells facilitate the homeostatic formation of TCM through the IL-15-pSTAT5-Eomes axis. (**A**) Expression levels of Eomes in TCM of CD8^+^ T cells were expressed via mean fluorescence intensity (MFI). (**B**) Phospho-flow cytometry of STAT5 in TCM of CD8^+^ T cells from CD1d^+/−^ and CD1d^−/−^ splenocytes’ response to IL-15 stimulation. The histogram illustrates the MFI of STAT5 phosphorylation in TCM. The dashed lines represent cells that are not stimulated by IL-15. (**C**) The expression of Eomes in CD62L^+^ CD122^+^ gated on TCM of CD8^+^ T cells from CD1d^+/−^ and CD1d^−/−^ mice was displayed and expression levels of Eomes were compared through MFI. Each dot represents a mouse. Four mice per group. The data are representative of three independent experiments. Means with SEM are shown. Statistics: non-paired two-tailed Student’s test: * *p* < 0.05, ** *p* < 0.01.

## Data Availability

The data that support the findings of this study are available on request from the corresponding author Yingyu Qin.

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
