# Peer review of "CD1d-Restricted NKT Cells Promote Central Memory CD8+ T Cell Formation via an IL-15-pSTAT5-Eomes Axis in a Pathogen-Exposed Environment"

_ijms, 2025, doi:10.3390/ijms26157272_

Round 1
Reviewer 1 Report
Comments and Suggestions for Authors
Long-lived memory T cells are essential for controlling persistent infections and eliminating cancers. However, the mechanisms underlying their generation and persistence remain poorly understood. In this study, the authors propose that CD1d-restricted NKT cells play a crucial role in the generation and maintenance of central memory CD8+ T cells through IL-15 signaling under non-SPF housing conditions. While these findings are potentially important, several issues must be addressed to strengthen the scientific conclusions.
(1) The authors used mice housed under non-SPF conditions and described “dirty” mice in the Introduction (https://doi.org/10.4049/jimmunol.1700453; https://doi.org/10.4049/jimmunol.2000171). In the Materials and Methods section, the authors stated that “The mice that were not highlighted as housed in SPF were housed with open cages in conventional animal facilities of Southeast University. (Line 248)”. For the sake of reproducibility, please provide more detailed information about the conventional mice used in this study. Additionally, please clarify what is meant by a “pathogen exposed environment”.
(2) Can adoptively transferred NKT cells from CD1d+/- mice restore the defective phenotype of CD44+CD62L+CD8+ T cells in CD1d-/- mice ?
(3) Please show the levels of IL-15Ralpha in CD8+ T cells from CD1d+/- and CD1d-/- mice.
(4) The authors described in the text that “Mechanistically, CD1d-restricted NKT cells elevated CD4+ T cell-derived IL-15 in CD1d+/- mice, activating the IL-15/pSTAT5/Eomes axis essential for TCM maintenance. (Line 19)”, that “Steady-state analysis revealed elevated IL-15 levels in CD1d+/- mice compared to CD1d-/- controls, primarily sourced from CD4+ T helper cells. (Line 80)”, and that “CD1d-restricted NKT cells may regulate IL-15 production through a CD4+ T cell-dependent mechanism, potentially contributing to enhanced survival and homeostatic proliferation of CD8+ TCM cells in D1d-/- mice. (Line 164)”. Please show data on IL-15 expression in CD4+ T cells. Under normal conditions, IL-15 is constitutively expressed by a wide variety of cell types and tissues, including monocytes, macrophages, dendritic cells, and fibroblasts.
Minor: Please check the accuracy of the descriptions below.
Line 56
help cell
Line 47
CD4+ T help cells
Line 62
However, the synthetic lipid antigen not can fully encompass the authentic physiological function of NKT cells.
Line 97
Interestingly, an opposite result was obtained: CD1d+/- mice had reduced TCM(CD44+CD62+) and TEM (CD44-CD62L+) populations compared to CD1d-/- mice (Fig. 1C).
Line 119
Underlying CD8 T cell differentiation and function are transcription factors.
Comments on the Quality of English LanguageThere are many typographical errors that should be corrected throughout the manuscript.
Author Response
For research article
|
Response to Reviewer 1 Comments
|
||
|
1. Summary |
|
|
|
Thank you very much for taking the time to review this manuscript. Please find the detailed responses below and the corresponding revisions/corrections highlighted/in track changes in the re-submitted files. |
||
|
2. Questions for General Evaluation |
Reviewer’s Evaluation |
Response and Revisions |
|
Does the introduction provide sufficient background and include all relevant references? |
yes |
[] |
|
Are all the cited references relevant to the research? |
Can be improved |
|
|
Is the research design appropriate? |
Can be improved |
|
|
Are the methods adequately described? |
Must be improved |
|
|
Are the results clearly presented? |
Can be improved |
|
|
Are the conclusions supported by the results? |
Can be improved |
|
|
3. Point-by-point response to Comments and Suggestions for Authors |
||
|
Comments 1: [The authors used mice housed under non-SPF conditions and described “dirty” mice in the Introduction (https://doi.org/10.4049/jimmunol.1700453; https://doi.org/10.4049/jimmunol.2000171). In the Materials and Methods section, the authors stated that “The mice that were not highlighted as housed in SPF were housed with open cages in conventional animal facilities of Southeast University. (Line 248)”. For the sake of reproducibility, please provide more detailed information about the conventional mice used in this study. Additionally, please clarify what is meant by a “pathogen exposed environment] |
||
|
Response 1: [We appreciate the reviewer's attention to the details of our animal housing conditions, which is crucial for reproducibility. Therefore, we revised the text: The mice that were not highlighted as housed in SPF were housed with open cages in conventional animal facilities of South east University that do not have air cleanliness standards, but they maintain regular ventilation (8 times/hour) and control room temperature within the range of 24℃, in accordance with the Chinese standard for laboratory animal environment and housing facilities (GB 14925-2023). (line 267)
|
||
|
Comments 2: [Can adoptively transferred NKT cells from CD1d+/- mice restore the defective phenotype of CD44+CD62L+CD8+ T cells in CD1d-/- mice ?] |
||
|
Response 2: We thank the reviewer for raising this critical question regarding functional validation of NKT cells. This part of experiment was added in Figure 1E. Figure legend: E. 4×105 NKT cells isolated from CD1d+/− mice were transferred into CD1d−/− mice. After 4 weeks of transfer, the frequencies of CD8+ TCM in the PBL, spleens and inguinal lymph nodes (ILNs) were analyzed. Non-cell transfer group (n=7), NKT cell transfer group (n=6). (line 446) Result despcription: Finally, a rescued experiment was processed using adoptive NKT cell transfer into CD1d−/− mice. Compared to the non-cell transfer group, slightly elevated frequencies of TCM were observed in the peripheral blood and inguinal lymphoid tissues in the NKT cell transfer group (Fig.1E). The results indicate that NKT cells promoting CD8+ TCM formation are not crucially dependent on CD1d molecules.(line 114) Materials and Method For iNKT cell transfer, iNKT cells were isolated from splenocytes and liver lymphocytes that were enriched via gradient centrifugation (40% Percoll, 600 x g, 15 min, RT) from CD1d+/− mice using the NK1.1+ iNKT cell isolation kit (Milteny Biotec, # 130-096-513 ). Subsequently, 4×105 isolated iNKT cells were intravenously injected into CD1d−/− mice. (line 316)
Comments 3: [Please show the levels of IL-15Ralpha in CD8+ T cells from CD1d+/- and CD1d-/- mice.] |
||
Response 3: We agree the reviewer's request for direct data on IL-15Rα levels. The data was shown in Figure 4D.
Result despcription: Notably, CD1d+/− mice exhibited significantly higher IL-15Rα levels on CD4+ T help cells compared to CD1d−/− mice, whereas CD8+ T cells showed comparable receptor expression in both groups (Fig. 4D).(line 177)
Comments 4: [The authors described in the text that “Mechanistically, CD1d-restricted NKT cells elevated CD4+ T cell-derived IL-15 in CD1d+/- mice, activating the IL-15/pSTAT5/Eomes axis essential for TCM maintenance. (Line 19)”, that “Steady-state analysis revealed elevated IL-15 levels in CD1d+/- mice compared to CD1d-/- controls, primarily sourced from CD4+ T helper cells. (Line 80)”, and that “CD1d-restricted NKT cells may regulate IL-15 production through a CD4+ T cell-dependent mechanism, potentially contributing to enhanced survival and homeostatic proliferation of CD8+ TCM cells in CD1d-/- mice. (Line 164)”. Please show data on IL-15 expression in CD4+ T cells. Under normal conditions, IL-15 is constitutively expressed by a wide variety of cell types and tissues, including monocytes, macrophages, dendritic cells, and fibroblasts.]
Response 4: We appreciate the reviewer's insightful comment regarding IL-15 detection in CD4+ T cells. We tried to detect IL-15 expression in CD4+T cells using flow cytometry with anti-IL-15 antibodies, but the signal we observed was consistently low. Since most commercially available fluorescently conjugated anti-IL-15 antibodies are designed for human targets, while our study focuses on mouse models, we opted to use a secondary antibody detection system instead. However, this approach may need further refinement for more reliable murine IL-15 detection.
As established in the literature, IL-15 signaling fundamentally differs from IL-2:
- IL-15 binds with high affinity to IL-15Rα intracellularly, forming a stable complex that traffics to the cell surface.
- Surface IL-15/IL-15Rα complexes then trans-present IL-15 to target cells expressing CD122/γc (e.g., memory CD8⁺ T cells), constituting the dominant physiological mode of IL-15 signaling.
Nolz JC, Richer MJ. Control of memory CD8+ T cell longevity and effector functions by IL-15. Mol Immunol. 2020;117:180-188. doi:10.1016/j.molimm.2019.11.011
- Critically, IL-15Rα deficiency phenocopies IL-15 deficiency, confirming their functional interdependence.
Kennedy MK, Glaccum M, Brown SN, Butz EA, Viney JL, Embers M, et al. Reversible defects in natural killer and memory CD8 T cell lineages in interleukin 15-deficient mice. J Exp Med. 2000;191:771–80. doi: 10.1084/jem.191.5.771
In our study, we noticed higher levels of systemic IL-15 (both mRNA and protein), along with a notable increase in IL-15Rα expression specifically on CD4+ T cells in CD1d⁺/⁻ mice compared to CD1d⁻/⁻ mice. Although directly detecting membrane-bound IL-15 on CD4+ T cells presented technical challenges due to the limited availability of effective murine-reactive antibodies, the simultaneous upregulation of both IL-15 ligand and its high-affinity receptor IL-15Rα on CD4⁺ T cells—consistent with the well-established trans-presentation mechanism—suggests a stronger potential for IL-15 delivery. With this in mind, we have carefully updated the relevant sections of the manuscript to reflect these observations and considerations.
In abstract:
Mechanistically, CD1d-restricted NKT cells elevated CD4⁺ T cell-derived IL-15 in CD1d⁺/⁻ mice, thereby activating the IL-15/pSTAT5/Eomes axis critical for TCM maintenance. Mechanistically, CD1d-restricted NKT cells increased IL-15Rα expression on CD4⁺ T cells in CD1d⁺/⁻ mice, potentiating IL-15 trans-presentation and thereby activating the IL-15/pSTAT5/Eomes axis critical for TCM maintenance.(line 21)
Steady-state analysis revealed elevated IL-15 levels in CD1d+/− mice compared to CD1d−/− controls, primarily sourced from CD4+ T helper cells. Under steady-state conditions, CD1d⁺/⁻ mice exhibited significantly elevated IL-15 (mRNA and protein) levels and increased IL-15Rα expression on CD4⁺ T cells compared to CD1d⁻/⁻ mice, demonstrating that CD1d-restricted NKT cells potentiate IL-15 trans-presentation capability, likely mediated by CD4⁺ T cells.(line 86)
Collectively, these findings suggest that CD1d-restricted NKT cells may regulate IL-15 production through a CD4+ T cell-dependent mechanism, Collectively, these findings suggest that CD1d-restricted NKT cells contribute to an immunological milieu characterized by enhanced IL-15 availability and augmented potential for IL-15 trans-presentation, particularly by CD4⁺ T cells due to their heightened IL-15Rα expression, which potentially contributes to enhanced survival and homeostatic proliferation of CD8+ TCM cells in CD1d−/− mice. (line 182)
Minor: Please check the accuracy of the descriptions below.
Line 56: help cell
Response: We thank the reviewer for catching this typo:
help cell helper cell (line 49)
Line 47: CD4+ T help cells
Response: We thank the reviewer for catching this typo:
CD4+ T help helper cells (line 51)
Line 62: However, the synthetic lipid antigen not can fully encompass the authentic physiological function of NKT cells.
Response: We appreciate the reviewer pointing out the need for clarity in this sentence.
However, the synthetic lipid antigen not can fully encompass the authentic physiological function of NKT cells. However, NKT cells activated by the synthetic lipid antigens cannot fully represent the true physiological functions of NKT cells, as these compounds induce non-physiological hyperactivation and cytokine profiles.(line 67)
Line 97: Interestingly, an opposite result was obtained: CD1d+/- mice had reduced TCM(CD44+CD62+) and TEM (CD44-CD62L+) populations compared to CD1d-/- mice (Fig. 1C).
Response: We thank the reviewer for highlighting the need to clarify the contrasting conditions:
Interestingly, an opposite result was obtained: CD1d+/− mice had reduced TCM (CD44+ CD62+) and TEM (CD44− CD62L+) populations compared to CD1d−/− mice (Fig. 1C). Interestingly, CD1d+/− mice showed decreased TCM (CD44+ CD62L+) and TEM (CD44− CD62L+) populations compared to CD1d−/− mice, which was opposite to results under antigen exposure conditions (Fig. 1C). (line 107)
Line 119:Underlying CD8 T cell differentiation and function are transcription factors.
Response: We thank the reviewer for noting the grammatical and conceptual issue:
Underlying CD8+ T cell differentiation and function are transcription factors. Underlying CD8+ T cell differentiation and function are regulated by transcription factors. (line 134)
Comments on the Quality of English Language
There are many typographical errors that should be corrected throughout the manuscript.
Response: We sincerely thank the reviewer for his/her insightful and constructive feedback, which has greatly strengthened the rigor and clarity of this manuscript. We have carefully addressed all points raised and incorporated the suggested revisions throughout the text and figures. Should any additional refinements be needed, we welcome further guidance and stand ready to implement necessary corrections promptly.
Reviewer 2 Report
Comments and Suggestions for Authors
In their manuscript, entitled CD1d-Restricted NKT Cells Promote Central Memory CD8+ T 2 Cell Formation via an IL-15-pSTAT5-Eomes Axis in a Pathogen-3 Exposed Environment, describe the function of NKT Cells in central memory formation. In the manuscript, there is a discrepancy between the results stated in the abstract, end of introduction part and the results part. In the results part, the authors describe that under SPF conditions there are reduced frequencies of central memory CD8 cells in CD1d-KO and J-alpha-KO mice, whereas under non-SPF conditions the opposite is the case. Here, under SPF conditions, NKT cells augment central memory T cell differentiation, whereas under non SPF conditions, NKT cells seem to be detrimental. The title suggests that the SPF results are described, also Figure 1 seems to only describe the SPF data. Abstract and last paragraph of the introduction describe the non-SPF conditions but the conclusions were drawn according to the SPF data. The data shown and described must be harmonized and the manuscript must be heavily revised. In addition, the role of NK cells during central memory T cell formation is already well described, as also stated by the authors, the focus should be rather directed into the SPF vs non-SPF issue describing the negative role of NKT cells in a dirty environment. This is already stated in the discussion (lane 195ff) and should also be underscored with data shown and discussed. Further, how did the microbiome differ between SPF and conventional housing? What was the status of the immune system (immune cell populations) in SPF vs conventional housed mice?
Author Response
For research article
|
Response to Reviewer 2 Comments
|
||
|
1. Summary |
|
|
|
Thank you very much for taking the time to review this manuscript. Please find the detailed responses below and the corresponding revisions/corrections highlighted/in track changes in the re-submitted files. |
||
|
2. Questions for General Evaluation |
Reviewer’s Evaluation |
Response and Revisions |
|
Does the introduction provide sufficient background and include all relevant references? |
Must be improved |
[] |
|
Are all the cited references relevant to the research? |
Must be improved |
|
|
Is the research design appropriate? |
Must be improved |
|
|
Are the methods adequately described? |
Must be improved |
|
|
Are the results clearly presented? |
Must be improved |
|
|
Are the conclusions supported by the results? |
Can be improved |
|
|
3. Point-by-point response to Comments and Suggestions for Authors |
||
|
Comments 1: [In their manuscript, entitled CD1d-Restricted NKT Cells Promote Central Memory CD8+ T 2 Cell Formation via an IL-15-pSTAT5-Eomes Axis in a Pathogen-3 Exposed Environment, describe the function of NKT Cells in central memory formation. In the manuscript, there is a discrepancy between the results stated in the abstract, end of introduction part and the results part. In the results part, the authors describe that under SPF conditions there are reduced frequencies of central memory CD8 cells in CD1d-KO and J-alpha-KO mice, whereas under non-SPF conditions the opposite is the case. Here, under SPF conditions, NKT cells augment central memory T cell differentiation, whereas under non SPF conditions, NKT cells seem to be detrimental. The title suggests that the SPF results are described, also Figure 1 seems to only describe the SPF data. Abstract and last paragraph of the introduction describe the non-SPF conditions but the conclusions were drawn according to the SPF data. The data shown and described must be harmonized and the manuscript must be heavily revised. In addition, the role of NK cells during central memory T cell formation is already well described, as also stated by the authors, the focus should be rather directed into the SPF vs non-SPF issue describing the negative role of NKT cells in a dirty environment. This is already stated in the discussion (lane 195ff) and should also be underscored with data shown and discussed. Further, how did the microbiome differ between SPF and conventional housing? What was the status of the immune system (immune cell populations) in SPF vs conventional housed mice?] |
||
|
Response 1: [We sincerely appreciate the reviewer's meticulous attention to the contextual nuances of our data. We acknowledge that the interplay between housing conditions and NKT cell function requires explicit clarification, and we apologize for any ambiguity in the original manuscript. Below we address the concerns point-by-point:
1. Title & Focus & data Alignment: The title "CD1d-Restricted NKT Cells Promote Central Memory CD8+ T Cell Formation via an IL-15-pSTAT5-Eomes Axis in a Pathogen-Exposed Environment" specifically highlights our core discovery in non-SPF (pathogen-exposed) settings (Fig. 1A, B, D, E). We agree that adding "non-SPF" and “In SPF” to the relevant figures would enhance precision and have revised them. Figs. 1 A, B, D, E (explicitly labeled "Non-SPF"): Demonstrate NKT-dependent TCM promotion under non-SPF conditions (CD1d+/−/Jα18+/− > CD1d−/−/Jα18−/−). Fig. 1C (explicitly labeled "In SPF"): an opposite result suggests that NKT cells inhibit TCM formation revealing the context-dependent role of NKT cells (TCM frequencies: CD1d−/− > CD1d+/− in SPF). 2. Contextualizing Opposite SPF Phenotypes: In antigen exposure condition (non-SPF): NKT cells promote TCM formation (mainly focused in this study) In SPF condition: NKT cells inhibit TCM formation We emphasize that the SPF data (Fig. 1C) do not contradict but rather contextualize our central hypothesis: NKT cells act as environmental sensors that promote TCM differentiation in pathogen-experienced milieus (non-SPF) while constraining memory cell accumulation in sterile settings (SPF). As discussed: Such a mechanism positions NKT cells as environmental rheostats, dynamically calibrating memory T cell reserves to match historical pathogen encounters. This aligns with evolutionary demands for immune systems to optimize resource allocation: in pathogen-rich settings, sustained NKT-mediated TCM support ensures rapid recall responses, whereas sterile environments may prioritize effector readiness over memory longevity. (line224)
|
||
|
Comments 2: [In addition, the role of NK cells during central memory T cell formation is already well described, as also stated by the authors, the focus should be rather directed into the SPF vs non-SPF issue describing the negative role of NKT cells in a dirty environment.] |
||
|
Response 2: We sincerely appreciate the reviewer's insightful comment regarding the established role of NKT cells in CD8⁺T cell memory formation and the valuable suggestion to emphasize our findings on NKT cell function in non-SPF environments.
As the reviewer rightly notes, foundational studies by Reilly et al. (PLoS One, 2012), Ueda et al. (Int Immunol, 2006), and Valente et al. (PNAS, 2019) have demonstrated that α-GalCer-activated iNKT cells promote CD8⁺T cell memory differentiation ( studied in SPF condition). However, as we highlighted in the Introduction (Lines 67): However, NKT cells activated by the synthetic lipid antigens cannot fully represent the true physiological functions of NKT cells, as these compounds induce non-physiological hyperactivation and cytokine profiles. Our study deliberately avoided α-GalCer stimulation to address a critical knowledge gap: We investigated the role of NKT cells in physiologically relevant microbial environments (non-SPF conditions). This revealed a previously unrecognized facet of NKT biology: Additionally, in steady state (without α-GalCer treatment) NKT cells inhibit TCM formation rather than promote TCM formation in SPF conditions [1]. Emma C. Reilly, E.A.T., Sandrine Aspeslagh,Jack R. Wands,Dirk Elewaut,Laurent Brossay, Activated iNKT cells promote memory CD8+ T cell differentiation during viral infection. Plos One, 2012. 7. [2]. Naoko Ueda, H.K., Daisuke Kamimura, Shinichiro Sawa, Kenichiro Seino, Takuya Tashiro, et al, CD1d-restricted NKT cell activation enhanced homeostatic proliferation of CD8+ T cells in a manner dependent on IL-4. International Immunology, 2006. 18: p. 1397-1404. [3]. M Valente, Y.D., E van Dinther, L Vimeux, M Fallet, V Feuillet, C G Figdor, Cross-talk between iNKT cells and CD8 T cells in the spleen requires the IL-4/CCL17 axis for the generation of short-lived effector cells. PNAS, 2019. 116: p. 25816-25827. All work used α-GalCer to activate iNKT cells and was studied in SPF condition.
Comments 3: [Further, how did the microbiome differ between SPF and conventional housing? What was the status of the immune system (immune cell populations) in SPF vs conventional housed mice?] |
||
Response 3: We sincerely thank the reviewer for raising these critical questions regarding microbiome and immune status differences between housing conditions. Below we provide detailed clarifications:
Microbiome Differences:
In this study, SPF conditions refer to mice housed in sterile isolators with filtered air, autoclaved food/water, and exclusion of defined pathogens (e.g., MHV, MNV, Helicobacter spp.). This environment minimizes microbial-driven immune activation, serving as a controlled baseline. Conversely, in the non-SPF (conventional) conditions, mice in open cages with ambient airflow were exposed to complex polymicrobial communities from personnel, equipment, and environmental sources (e.g., Enterobacteriaceae, Clostridiales, Bacteroidetes). This replicates natural pathogen encounters, enabling analysis of physiologically relevant immune memory formation.
We further described the non-SPF housing condition (conventional animal facilities) in the section of Materials and Methods as shown below:
The mice that were not highlighted as housed in SPF were housed with open cages in conventional animal facilities of South east University that do not have air cleanliness standards, but they maintain regular ventilation (8 times/hour) and control room temperature within the range of 24℃, in accordance with the Chinese standard for laboratory animal environment and housing facilities (GB 14925-2023). (line 267)
We sincerely thank the reviewer for his/her insightful and constructive feedback, which has greatly strengthened the rigor and clarity of this manuscript. We have carefully addressed all points raised and incorporated the suggested revisions throughout the text and figures. Should any additional refinements be needed, we welcome further guidance and stand ready to implement necessary corrections promptly.
Round 2
Reviewer 1 Report
Comments and Suggestions for Authors
The authors have successfully addressed all my comments and concerns.
The FACS plots in Figure 1E should be labeled with “No transfer” and “NKT transfer” for clarity.
Author Response
Comment 1:
The FACS plots in Figure 1E should be labeled with “No transfer” and “NKT transfer” for clarity.
Response 1:
We sincerely apologize for the oversight in labeling the FACS plots in Figure 1E. We have revised Figure 1E to include the specific labels.
Reviewer 2 Report
Comments and Suggestions for Authors
the quality of the manuscript has increased significantly now. Also it is better to read and to understand. However, it is necessary to put more emphasis on the main focus, that an microbe rich environment is needed for NKT dependent T memory formation and write this even more clearly. this should be better assessed in the abstract and introduction. In addition, in the results part, lane 99 "conventional conditions" should be exchanged to non-SPF since the term conventional condition could be interpreted as SPF.
Author Response
Comment 1: the quality of the manuscript has increased significantly now. Also it is better to read and to understand. However, it is necessary to put more emphasis on the main focus, that an microbe rich environment is needed for NKT dependent T memory formation and write this even more clearly. this should be better assessed in the abstract and introduction.
Response 1: We sincerely appreciate the reviewer pointing this out. We agree with the reviewer and have made the revision accordingly. Please see the changes detailed below:
Abstract:
This study reveals that CD1d-restricted NKT cells regulate central memory T cell (TCM) generation exclusively in a microbe-rich ("dirty") environment. Under non-SPF housing, CD1d⁺/⁻ and Ja18⁺/⁻ mice exhibited enhanced TCM formation compared to NKT-deficient controls (CD1d⁻/⁻/Ja18⁻/⁻), demonstrating that microbial experience is required for NKT-mediated TCM regulation.
This study demonstrates that under non-SPF housing, CD1d⁺/⁻ and Ja18⁺/⁻ mice exhibit enhanced central memory T cell (TCM) formation compared to CD1d-restricted NKT cell-deficient controls (CD1d⁻/⁻/Ja18⁻/⁻). This implicates CD1d-restricted NKT cells in regulating TCM development.
Introduction
However, NKT cells activated by the synthetic lipid antigens cannot fully represent the true physiological functions of NKT cells, as these compounds induce non-physiological hyperactivation and cytokine profiles. Importantly, physiological iNKT activation occurs via microbial-triggered cytokine receptors (e.g., IL-12R/IL-18R)[15], highlighting the necessity of pathogen exposure for studying their endogenous functions. Our prior work showed iNKT cells enhance CD8⁺ T cell effector function without exogenous antigens [16], yet their role in memory differentiation under physiologically relevant (microbe-rich) conditions remains unknown.
Given that activation can occur through the constitutive expression of inflammatory cytokine receptors, such as IL-12R and IL-18R, even in the absence of TCR signals, this enables direct or indirect microbial-induced activation of NKT cells [15]. Additionally, our previous study has reported that iNKT cells can enhance the effector function of CD8+ T cells through direct interaction without exogenous antigens, thereby promoting CD8+ T cell-mediated antitumor activity [16]. However, the impact of CD1d-restricted NKT cells on memory CD8+ T cell differentiation in the absence of synthetic lipid antigen stimulation remains to be elucidated.
In this study, we resolved this gap using non-SPF ("dirty") housing to model natural microbial exposure. We demonstrate that CD1d-restricted NKT cells drive central memory T cell (TCM) formation exclusively in a microbe-rich environment—a dependency absent in SPF settings. we observed higher frequencies of TCM in CD1d+/− or Ja18+/− compared to CD1d−/−mice or Ja18−/− (deficient in iNKT cells) mice within a “dirty” environment, suggesting the essential role of CD1d-restricted NKT in TCM formation.
Eomesodermin (Eomes) is a transcription factor crucial for the development and func-tion of CD8+ memory T cells [17]. In addition, IL-15 contributes to the homeostatic proliferation and survival of memory CD8+ T cells under steady state conditions [18]. Mechanistically, we observed that Eomes expression is strictly correlated with TCM formation. CD1d⁺/⁻ mice exhibited significantly elevated IL-15 (mRNA and protein) levels and increased IL-15Rα expression on CD4⁺ T cells compared to CD1d⁻/⁻ mice, demonstrating that CD1d-restricted NKT cells potentiate IL-15 trans-presentation capability, likely mediated by CD4⁺ T cells. Subsequent investigations established that CD1d-restricted NKT cells promote TCM formation through an IL-15-STAT5 phosphorylation-Eomes upregulation axis. Adoptive transfer experiments confirmed an NKT-dependent survival advantage for memory CD8⁺ T cells specifically in microbiota-experienced hosts.
Comment 2 :
In addition, in the results part, lane 99 "conventional conditions" should be exchanged to non-SPF since the term conventional condition could be interpreted as SPF.
Response 2: We sincerely appreciate the reviewer pointing this out. We agree with the reviewer and have made the revision accordingly. Please see the changes detailed below:
To investigate the physiological role of NKT cells in memory CD8+ T cell generation, we compared littermate CD1d+/− and CD1d−/− mice (lacking NKT cells) housed under non-SPF conditions conventional conditions. Without experimental intervention, both strains exhibited an age-dependent increase in central memory T cell (TCM) frequencies, but CD1d+/− mice consistently showed higher TCM frequencies than CD1d−/− mice from early to late ages (Fig. 1A, B). To determine whether CD1d-restricted NKT cells promote TCM formation specifically in a pathogen-exposed environment, we compared littermate CD1d+/− and CD1d−/− mice housed under SPF conditions.Interestingly, CD1d+/− mice showed decreased TCM (CD44+ CD62L+) and TEM (CD44− CD62L+) populations compared to CD1d−/− mice, which was opposite to results under antigen exposure conditions (Fig. 1C). Since CD1d deficiency depletes all CD1d-restricted NKT cells, we further assessed Jα18−/− mice, which lack only iNKT cells. Consistent with our earlier findings, Jα18−/− mice had reduced TCM populations compared to Jα18+/− controls across all ages (Fig. 1 D). However, the complete ablation of all CD1d-restricted NKT cells (CD1d−/−) led to more severe impairments in the formation of central memory T cells (TCMs) compared to the ablation of iNKT cells alone (Jα18−/−). Finally, a rescued experiment was processed using adoptive NKT cell transfer into CD1d−/− mice. Compared to the non-cell transfer group, slightly elevated frequencies of TCM were observed in the peripheral blood and inguinal lymphoid tissues in the NKT cell transfer group (Fig.1E). The results indicate that NKT cells promoting CD8+ TCM formation are not crucially dependent on CD1d molecules. Collectively, these results indicate that CD1d-restricted NKT cells promote TCM formation under non-SPF conditions conventional conditions.